# Invasive species allelopathy decreases plant growth and soil microbial activity

**Tongbao Qu**[1]*, **Xue Du**[1], **Yulan Peng**[1], **Weiqiang Guo**[1], **Chunli Zhao**[1],
**Gianalberto Losapio**[2]

**1** College of Horticulture, Jilin Agricultural University, Changchun, PR China, **2** Department of Biology,
Stanford University, Stanford, CA, United States of America

* qvtb@sina.com

**Data Availability Statement:** All relevant data are within the manuscript.

**Funding:** This work was supported by the Department of Science and Technology of Jilin Province (20190303078SF). GL was supported by

## Abstract

According to the 'novel weapons hypothesis', invasive success depends on harmful plant biochemicals, including allelopathic antimicrobial roots exudate that directly inhibit plant growth and soil microbial activity. However, the combination of direct and soil-mediated impacts of invasive plants via allelopathy remains poorly understood. Here, we addressed the allelopathic effects of an invasive plant species (*Rhus typhina*) on a cultivated plant (*Tagetes erecta*), soil properties and microbial communities. We grew *T. erecta* on soil samples at increasing concentrations of *R. typhina* root extracts and measured both plant growth and soil physiological profile with community-level physiological profiles (CLPP) using Biolog Eco-plates incubation. We found that *R. typhina* root extracts inhibit both plant growth and soil microbial activity. Plant height, Root length, soil organic carbon (SOC), total nitrogen (TN) and AWCD were significantly decreased with increasing root extract concentration, and plant above-ground biomass (AGB), below-ground biomass (BGB) and total biomass (TB) were significantly decreased at 10 mg·mL$^{-1}$ of root extracts. In particular, root extracts significantly reduced the carbon source utilization of carbohydrates, carboxylic acids and polymers, but enhanced phenolic acid. Redundancy analysis shows that soil pH, TN, SOC and EC were the major driving factors of soil microbial activity. Our results indicate that strong allelopathic impact of root extracts on plant growth and soil microbial activity by mimicking roots exudate, providing novel insights into the role of plant–soil microbe interactions in mediating invasion success.

## Introduction

Invasive species degrade the environment, reduce biodiversity and alter ecosystem processes, threatening livelihoods worldwide [1]. The mechanisms underlying successful plant invasion is a long-standing topic which is debated in ecology [2–5]. The theory of 'novel weapon hypothesis' (NWH) states that plant invasive success depends upon the ability to release novel phytochemicals into the invaded community [6, 7]. According to the NWH, invasive success depends on plant biochemical weapons, including allelopathic and antimicrobial root exudates, that directly inhibit native plant growth or mediate new plant–soil microbial

the Swiss National Science Foundation (P2ZHP3_187938).

**Competing interests:** The authors have declared that no competing interests exist.

interactions. Today, allelopathic effects of invasive species on co-occurring native plants are widely recognized as the major factors responsible for successful invasion in plant communities [8]. Despite the large number of previous studies investigating the allelopathic potential of invasive plants [9, 10], it is unclear about the role soil properties and microbial community play in mediating the impact of invasive species on plants. Addressing the effects of alien invasion on soil biota may improve our understanding of the key role soil organisms play in ecological systems, mediating biotic interactions.

It is important to notice that aboveground and belowground systems are not closed and isolated compartment, but they are rather linked together, exchanging energy and materials. Soil microbes play a crucial role in soil nutrient cycling and soil health [11]. Invasive plants can outcompete and displace native species thanks to allelopathy, which changes soil conditions via underground root exudates [12]. Invasive species root allelochemicals released into the soil can change soil properties, in turn affecting the composition and diversity of soil microbial community [13, 14]. Therefore, it is of great significance to study the effects of root allelochemicals from invasive plants on the soil physical and chemical properties and soil microorganisms in order to improve our understanding of plant invasion mechanisms. Yet, we have a limited knowledge of the relative contribution of direct and indirect allelopathic effects between invasive and other species.

*Rhus typhina* L. (Anacardiaceae) is a perennial shrub that make clones through production of underground root buds and forms large, dome-shaped canopies of a single genet, with progressively younger ramets developing around the periphery [15]. As an invasive shrub in most areas of China, *R. typhina* is classified as one of the most destructive invading exotic species and poses a great threat to terrestrial ecosystems due to his great ability to adapt and reproduce fast and its resistance to eradication once is established [16–18]. *Tagetes erecta* L. (Compositae) is a widespread ornamental plant and is commonly known as *Marigold*, native to Mexico [19]. It is found that there are patterns of negative effects grown in conjunction with garden trees where may co-occur with *R. typhina*, which could be allelopathic effects impacted by *R. typhina*. Recent studies highlight that *R. typhina* impacts both native plant communities and *T. erecta* by inhibiting seed germination and seedling growth [20, 21]. These evidences the potential allelopathy of *R. typhina* and its broad impact on biodiversity and local environmental conditions. At present, however, it remains unclear whether and how *R. typhina* impacts soil microorganisms through root extracts, in turn affecting plant growth.

Here, we examine the allelopathic effects of the invasive shrub *R. typhina* on the cultivated plant *T. erecta* via changes in soil conditions and microbial communities. We compare soil properties, metabolic activity and microbial carbon utilization at increasing concentration of *R. typhina* root extracts and examine the effects of changing soil conditions on *T. erecta* growth. We hypothesize that 1) *R. typhina* root extracts inhibit plant growth; 2) *R. typhina* root extracts decrease soil conditions; 3) The reasons of inhibition are the extracts negatively impact soil conditions and rhizophere soil microbial carbon source utilization.

## Materials and methods

### Root extracts of *R. typhina*

Healthy, fully developed and fresh roots of *R. typhina* were collected from Changchun (43˚52' N, 125˚21' E), China in August 2017. The region has a temperate continental humid climate with an average annual air temperature of 4.6˚C and annual precipitation ranges from 600 mm to 700 mm. Annual freezing period is 5 months with an average 140–150 frost-free days.

1000 g of roots were taken from the plants of *R. typhina* and washed, air dried for 48 h and then powdered in a grinder until 98% of the powder could pass through 10 mm mesh sieve. The samples were extracted with deionized water, shaken at 50 rpm$^{-1}$ for 48 h, then

concentrated by rotary evaporation, and finally dried in a freeze dryer. The obtained substance was 24.96 g, and the extraction rate was 2.5%. The concentration of the extract solution was set to 2.5, 5, 7.5 and 10 mg·mL$^{-1}$ according to previous experimental data, then kept in brown bottles at 4°C for subsequent processing [22].

## Experimental design and soil sample collection

Seeds of *T. erecta* were collected on the campus of Jilin Agricultural University, Changchun, China in October 2016. The selected seeds were soaked and disinfected with 0.3% KMnO$_4$ solution for 5 min, then rinsed with distilled water 3–5 times. We initially sowed ten seeds of *T. erecta* per pot (18 cm diameter×21 cm height) in a climate-controlled incubator at 25°C with diurnal lighting (light intensity was set at 2200 Lux on a 14:10 h light–dark cycle). After germination, we kept five plants in each pot. We use a mix of garden soil, turfy soil and perlite by volume 6: 3: 1. Pots were randomly assigned to treatments (i.e., control, 2.5, 5, 7.5 and 10 mg·mL$^{-1}$ of root extracts). Pots were watered with 20 mL of different concentrations of root extracts every day, or 20 mL of distilled water for the control groups. Each treatment was replicated three times, for a total of $n = 15$ pots and $n = 75$ plants.

After 60 days, plant and soil samples were collected for analyses. Soil samples were screened through sieves with 2 mm mesh and divided into two parts after removal of coarse roots and stones. One part was air dried and used for physico-chemical analyses whereas the other one used for EcoPlates (Biolog, Hayward, CA, USA) and microbial activity analyses. All samples were stored at 4°C before analysis.

## Growth of *T. erecta*

We measured plant height, root length, aboveground biomass (AGB) and underground biomass (BGB). AGB was measured by clipping standing plant materials to 1 cm above ground level.

## Soil physico-chemical properties

We measured the following soil properties: pH, electrical conductivity (EC), water content (SWC), soil organic carbon (SOC), total nitrogen (TN). The soil pH and EC of the samples were measured using a digital pH-meter (CP-401, ELMETRON) and electric conductivity meter (DDSJ-308A); SWC was measured by the oven-drying method; SOC was measured using a Mebius method by the Walkley-Black acid digestion; TN was determined with an auto-analyzer (Foss 2100, FOSS Kjeltec®) using the Kjeldahl method following vitriol digestion.

## Community level physiological profiles (CLPP)

Community-level physiological profiles (CLPP) is a good method used for measured microbial activity of rhizosphere soil, which were measured with Biolog Eco-plates$^{TM}$ (Biolog Inc., Hayward, CA, USA), which contains 31 types of carbon sources and one substrate-free control. Soil samples (10 g d.w.) were firstly shaken (5 min, 180 r·min$^{-1}$) after dilution with 90 mL of distilled sterile water (pH 7) for 20 min at 20°C. Then aliquot (150 μL) were inoculated into wells of 96 Biolog Eco-plates and incubated at 25°C. The rate of utilization was showed by the reduction of the tetrazolium, a redox indicator dye that changes from colorless into purple. Lastly, substrate utilization was monitored by measuring light absorbance at 590 and 750 nm. The first measurement was made immediately after inoculation, and the subsequent ones were read every 12 h for a total of 240 h. The readings for individual substrates were corrected for background absorbance by subtracting the absorbance of the control (water) well and one of the absorbance of the first reading.

Average well color development (AWCD) was determined as follows: AWCD = $\Sigma(C_{590} - C_{750})/31$. $C_{590}$ and $C_{750}$ are optical density value in 590 nm and 750 nm from each well, corrected subtracting the blank well (inoculated with distilled water) values from each plate well. Diversity indices of AWCD were calculated as Shannon index ($H = -\Sigma(Pi \times \ln Pi)$), Pielou evenness index ($E = H/H_{max} = H/\ln S$), and Simpson diversity index ($D = 1-\Sigma Pi^2$). $Pi = C_{590}-C_{750}/\Sigma(C590—C_{750})$, $S$ were calculated as the number of oxidized carbon substrates (Qu et al. 2020).

According to the change curve of carbon source utilization over time, incubation at 180 h was used to calculate AWCD, $H$, $E$ and $D$ since it was the highest average well color development (AWCD) that was most consistent between treatments. The categorized carbon substrates were grouped into six guilds–carbohydrates, amino acids, carboxylic acids, polymer, phenolic acids and amine.

## Statistical analysis

The experiment has been repeated three times. Differences in plant growth, soil physico-chemical properties and microbial activity between the different concentration extracts were tested by one-way ANOVA. Tukey's tests were conducted separately to for post hoc analysis of differences between specific factor levels. Redundancy analysis (RDA) was conducted to link the use of particular carbon substrates to soil properties and to compare CLPPs. Monte Carlo permutations (MCP) with 999 permutation were used to test the significance of different factors along the first axis.

Data analysis was done with the SAS Statistical Software and by Canoco.

## Results

### Growth of *T. erecta*

*R. typhina* root extracts significantly reduced, root length, AGB, BGB and TB of *T. erecta* compared to the control group at 10 mg mL$^{-1}$ (Table 1 and Fig 1). Plant height firstly increased then decreased, and root length significantly decreased with extract concentrations increasing (Table 1). There were no significant different AGB, BGB and TB among different concentration below 10 mg mL$^{-1}$. At highest concentration (10 mg mL$^{-1}$), plant height was reduced by 7.38% ($P < 0.05$), root length decreased by 13.13%, and plant AGB, BGB and TB significantly decreased by 27.7%, 22.9% and 26.5% than control, respectively ($P < 0.05$).

### Soil physicochemical properties

Root extracts significantly increased soil EC and pH, and significantly decreased SOC and TN (Table 2). No significant effect was found for SWC and C/N (Table 2). Soil pH increased with

**Table 1. Growth of *Tagetes erecta* plants in response to different concentrations of *Rhus typhina* root extract.**

| Concentration (mg·mL$^{-1}$) | Plant height (cm) | Root length (cm) | AGB (g) | BGB (g) | TB (g) |
|---|---|---|---|---|---|
| Control | 16.52±0.29 bc | 16.30±0.36 a | 18.72±1.99 a | 6.23±1.09 a | 24.95±3.47 a |
| 2.5 | 19.14±0.84 a | 15.32±0.62 bc | 17.87±1.02 a | 6.55±0.39 a | 24.42±1.34 a |
| 5 | 17.82±1.53 ab | 15.64±0.41 b | 16.86±1.03 a | 5.78±0.51 a | 22.44±0.54 a |
| 7.5 | 15.94±0.48 c | 15.20±0.30 bc | 16.73±1.23 a | 6.46±0.58 a | 23.20±0.69 a |
| 10 | 15.30±0.85 c | 14.16±0.54 c | 13.53±0.08 b | 4.80±0.22 b | 18.33±0.29 b |

*Note*. Value ± SE with ANOVA results ($n$ = 3). Different letters represent significant differences between different concentration ($P < 0.05$; Tukey's test). AGB: aboveground biomass, BGB: belowground biomass, TB: total biomass.

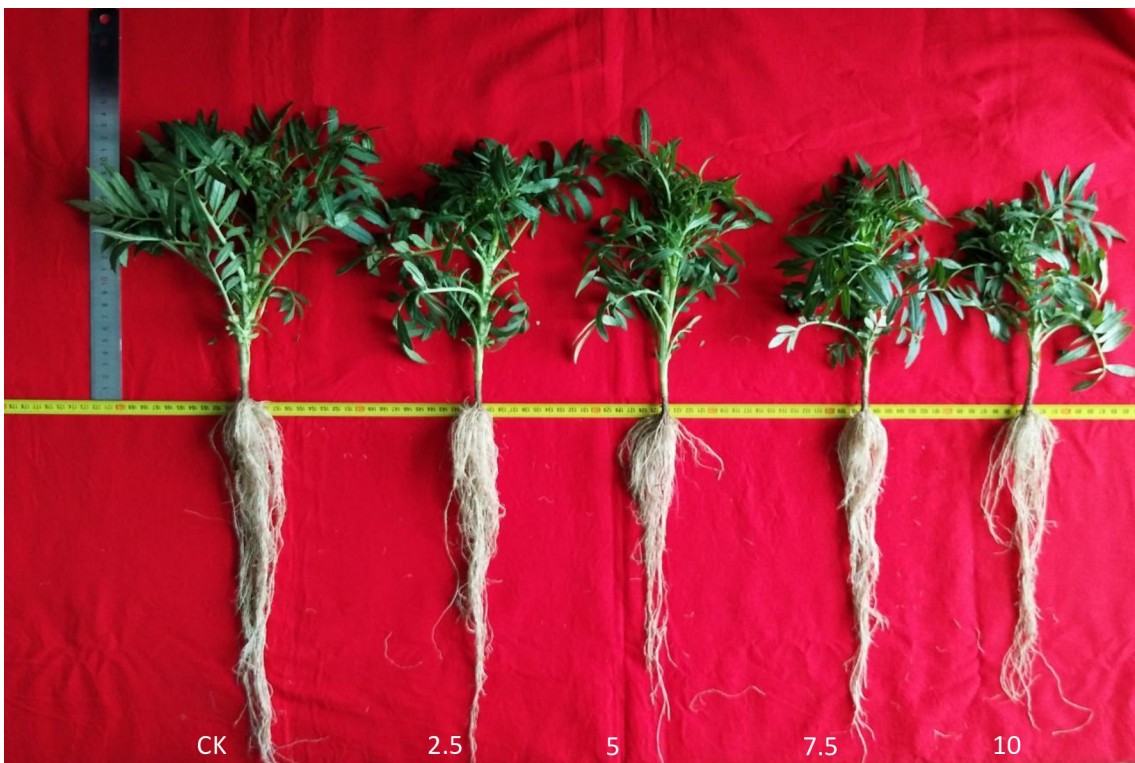

**Fig 1. Growth of *T. erecta* at different *R. typhina* root extract concentration after 60 days.** CK, 2.5, 5, 7.5, 10 represent different extract concentrations of 0, 2.5, 5, 7.5, 10 mg·mL$^{-1}$, respectively.

extract concentration increasing, and the highest pH and EC were at concentration of 10 mg·mL$^{-1}$ and 5 mg·mL$^{-1}$ with changes of 11.5% and 50.6%, respectively ($P < 0.05$). SOC and TN significantly decreased with increasing extract concentrations, and the lowest SOC and TN were at 10 mg·mL$^{-1}$ concentration, with average change of 25.4% and 22.6%, respectively ($P < 0.05$).

## Community level physiological profiles of soil microbial communities

After 12 h inoculation, a first substrate conversion was indicated by beginning color development in the microplate wells. AWCD then rapidly increased from 12–180 h incubation followed by clear differentiation among different treatments. After 180 h we observed a steady,

**Table 2. Soil physicochemical responses to different concentrations of *R. typhina* root extract.**

| Concentration | EC | pH | SWC | SOC | TN | C/N |
|---|---|---|---|---|---|---|
| (mg·mL$^{-1}$) | (µS·cm$^{-1}$) | | (%) | (g·kg-1) | (g·kg-1) | |
| Control | 86.73±0.25 e | 6.08±0.02 cd | 20.58± 1.67 a | 68.93±2.68 a | 5.22±0.11 a | 13.19±0.50 a |
| 2.5 | 94.30±0.10 d | 5.92±0.06 d | 16.43± 2.82 a | 59.17±0.51 b | 4.65±0.03 b | 12.72±0.18 a |
| 5 | 130.60±0.10a | 6.18±0.03bc | 16.01± 0.93a | 62.79±0.34 b | 4.95±0.11 a | 12.69±0.31 a |
| 7.5 | 127.70±0.36b | 6.32±0.01b | 16.73± 1.64a | 54.59±0.18 c | 4.34±0.17 c | 12.60±0.55 a |
| 10 | 99.27±0.15c | 6.78±0.13 a | 19.57±2.00 a | 51.44±1.72 c | 4.04±0.04 d | 12.75±0.35 a |

*Note.* Value ± SE with ANOVA results ($n = 3$). Different letters represent significant differences among different concentrations ($P < 0.05$; Tukey's test). EC: electrical conductivity, SWC: soil water content, TN: total nitrogen, SOC: soil organic carbon and C/N: C: N ratio.

saturating increase (Fig 2). Higher AWCD values were maintained in controls compared to all extract concentrations until the end of the incubation period of 240 h (Fig 2). At the peak (i.e., 180 h), results indicate that AWCD decreased by 15.20%, 5.71%, 23.46% and 28.56% at 2.5, 5, 7.5, and 10 mg·mL$^{-1}$ concentrations compared to control, respectively (all $P < 0.05$, Fig 2).

All 6 groups of carbon source were metabolized in Biolog Eco-plates in different concentrations of extracts (Fig 3), indicating different functional diversity of soil microbiota. The extracts from *R. typhina* roots significantly reduced the carbon source utilization of carbohydrates, carboxylic acids and polymers, and significantly enhanced phenolic acid (all $P < 0.05$) compared to control. Amino acids and amines firstly increased and then decreased with the increasing of extract concentrations ($P < 0.05$). The carbon source mostly used in controls was polymers and the least was phenolic acid, same patterns of response for phenolics acids than aminoacids. Among the treatments, phenolic acids were the most common, especially at 5 mg·mL$^{-1}$, while amines were the least used, especially at 10 mg·mL$^{-1}$ ($P < 0.05$).

## Diversity of soil microbial carbon utilization

Shannon index and Simpson diversity index firstly significantly increased and then significantly decreased and Pielou evenness index decreased gradually with the increasing of extract concentrations ($P < 0.05$) (Fig 4). Shannon index, Pielou evenness index and Simpson diversity index reached lowest values at 10 mg·mL$^{-1}$ extract, decreasing by 10.74%, 2.24% and 1.80% as compared to control, respectively ($P < 0.05$).

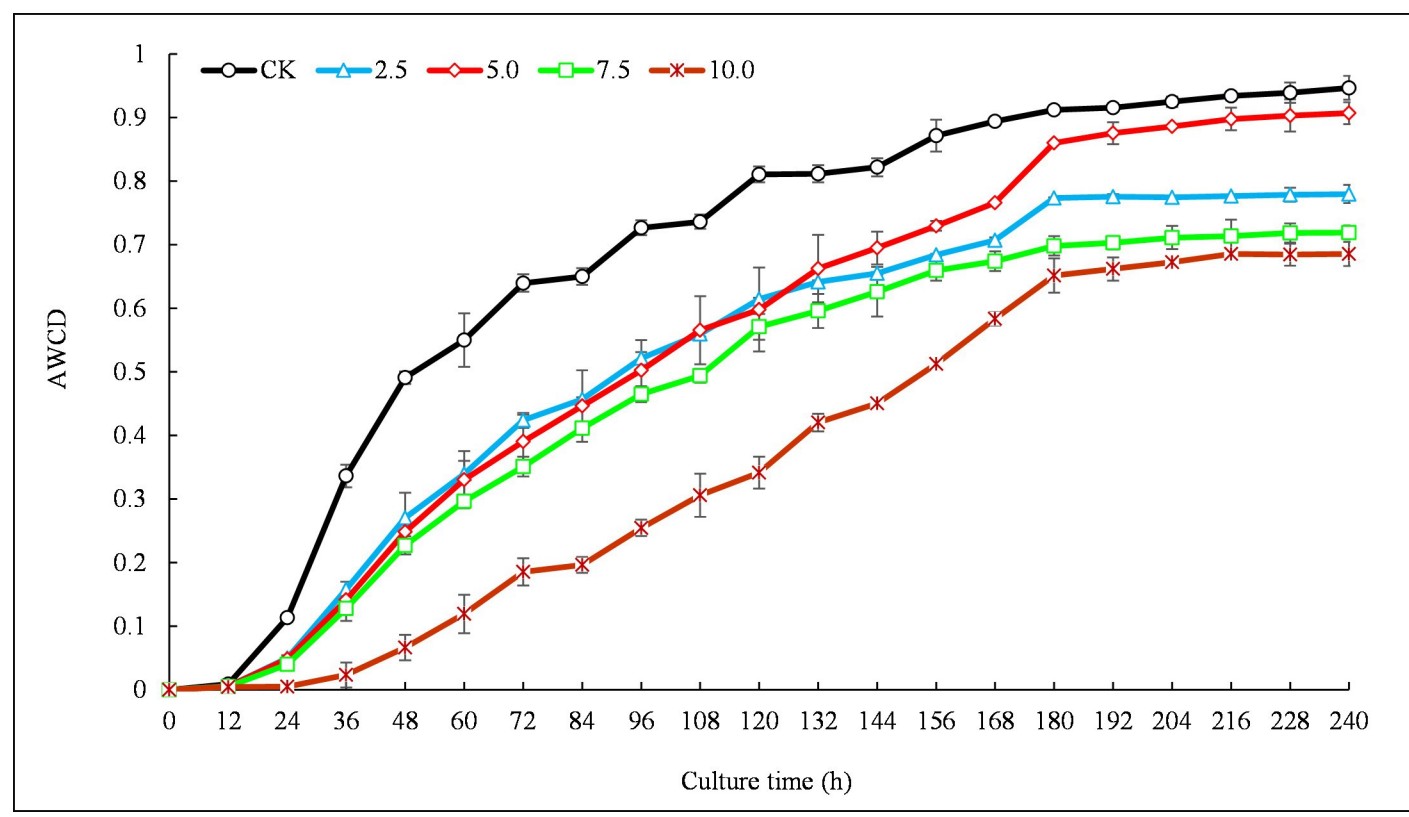

**Fig 2. Variation in average well color development (AWCD) of soil microbes at different *R. typhina* root extract concentration over time.** CK, 2.5, 5, 7.5, 10 represent different extract concentrations of 0, 2.5, 5, 7.5, 10 mg·mL$^{-1}$, respectively. Values represent means ± standard deviation.

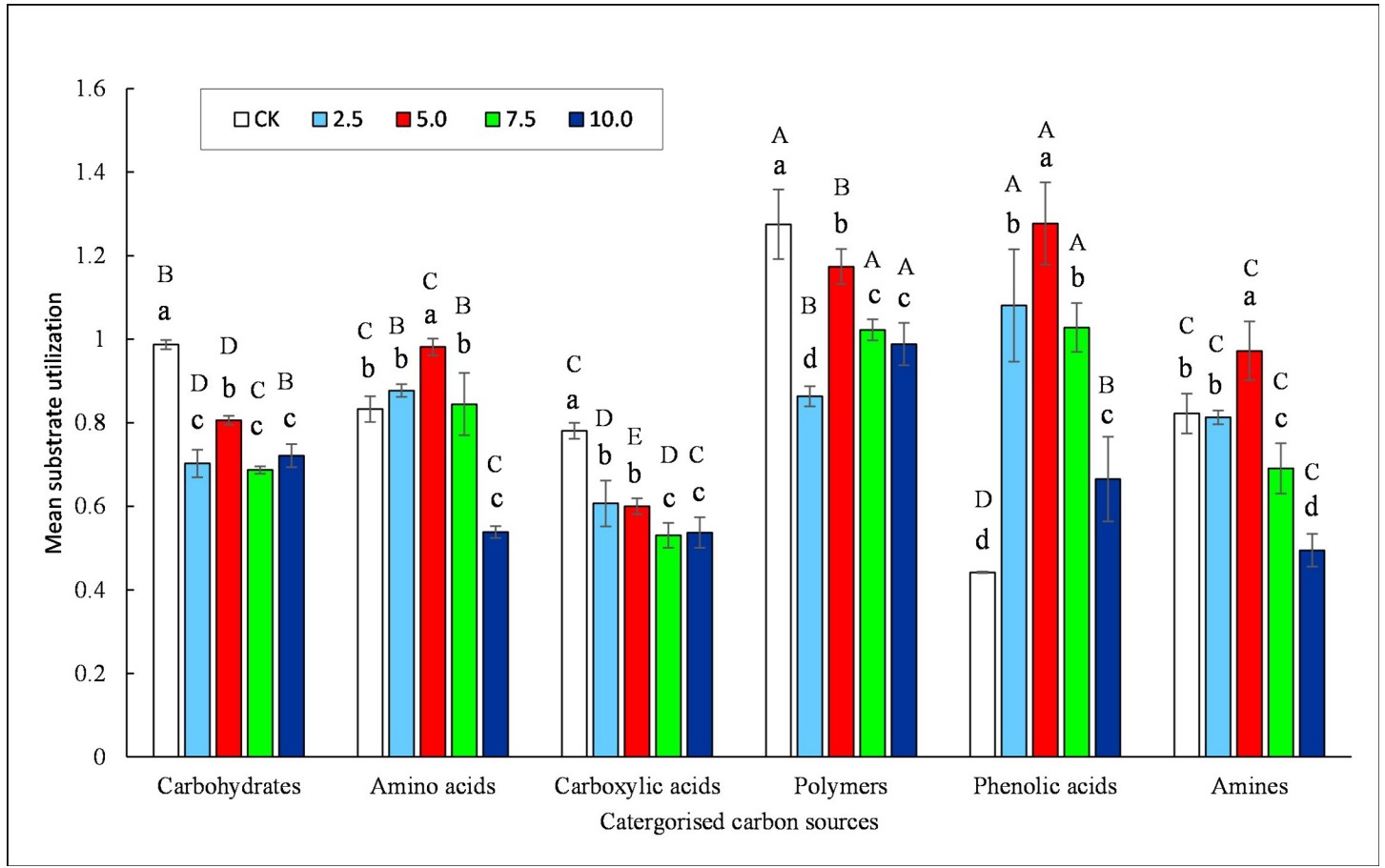

**Fig 3. Substrate utilization patterns of different carbon source by soil microbial community after 180 h of incubation in the Biolog Eco-plates at different *R. typhina* root extract concentrations.** Values represent means ± standard deviation (*n* = 15). Significant differences between carbon sources utilization are indicated with capital letters (A-C; *P* < 0.05); significant differences between treatments of concentration are indicated with lowercase letters (a-c; *P* < 0.05).

### Redundancy analysis (RDA) of soil carbon utilization

We report results of RDA in biplot graph with response variables (31 kinds of soil microbial carbon utilization) and explanatory variables (soil physicochemical characteristics) (Fig 5A). Eigenvalues for the first, second, third and fourth axes were 0.5201, 0.2927, 0.0345, respectively. Axis 1 significantly explained most of the variation in soil microbial carbon utilization (*P* <0.001). All the soil physicochemical characteristics could explain 85.20% of soil microbial carbon utilization variation (Monte Carlo permutation test with 999 permutation, *P* <0.001). Conditional effects show that the main factors driving soil microbial carbon utilization were pH (32.3%, *P* = 0.008), TN (29%, *P* = 0.002) and EC (19.7%, *P* = 0.004).

RDA analysis based on carbon sources and soil physicochemical characteristics under different concentrations of extracts was performed through the graph of triplots (Fig 5B). Eigenvalues for the first, second, third and fourth axes were 0.5792, 0.2411 and 0.0845, respectively. Axis 1 explained most of the variation in soil microbial carbon utilization (*P* = 0.002). All the soil data together could explain 91.9% of soil microbial carbon utilization variation (Monte Carlo permutation test with 999 permutation, *P* <0.001). Conditional effects showed that the main factors affecting soil microbial carbon utilization were pH (51.8%, *P* = 0.002), SOC (22.4%, *P* = 0.002) and EC (14%, *P* = 0.002). According to soil pH, the order of affecting soil

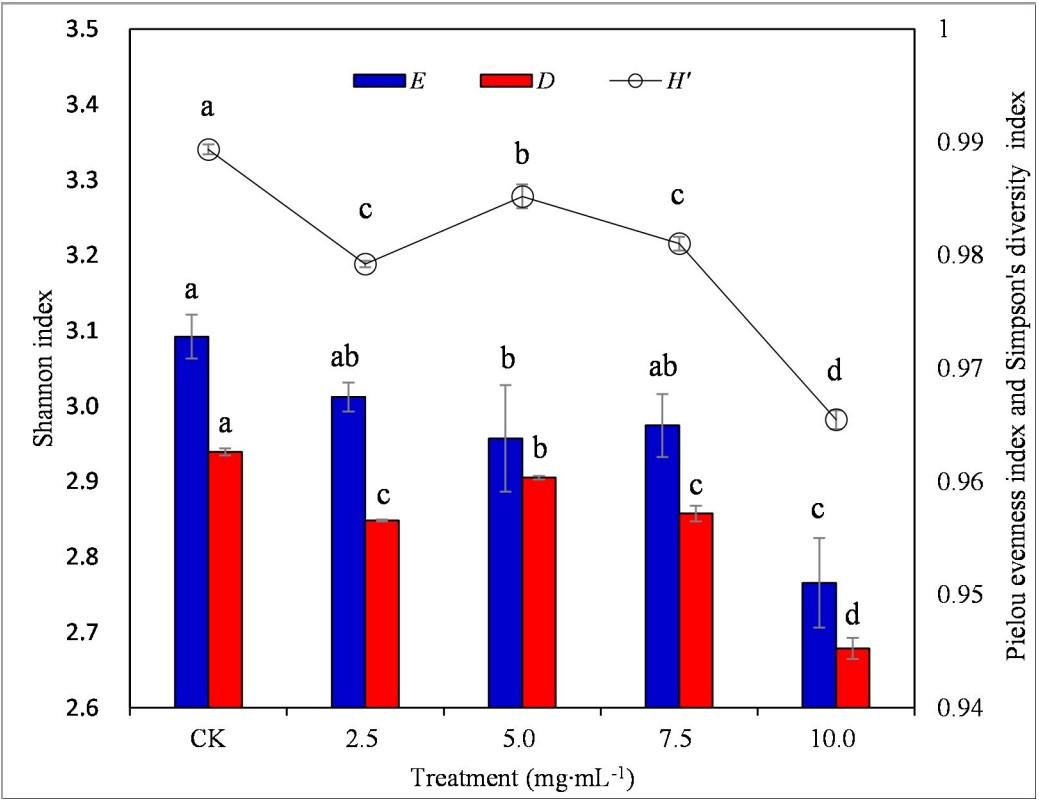

**Fig 4. Diversity and evenness indices of soil microbial communities.** *H'*: Shannon index; *D*: Simpson diversity index; *E*: Pielou evenness index. Values represent means ± standard deviation (*n* = 15). Significant differences between treatments are indicated with lowercase letters (a-c; *P* < 0.05).

microbial categorized carbon utilization by concentration of extracts was 10, 7.5, 2.5, CK and 5 mg·mL$^{-1}$. Results also showed that soil pH mainly decreased microbial utilization to Amino acids and Amines, and SOC mainly enhanced utilization of carboxylic acids, polymers and carbohydrates, and EC mainly positively affected utilization of phenolic acid (Fig 5B).

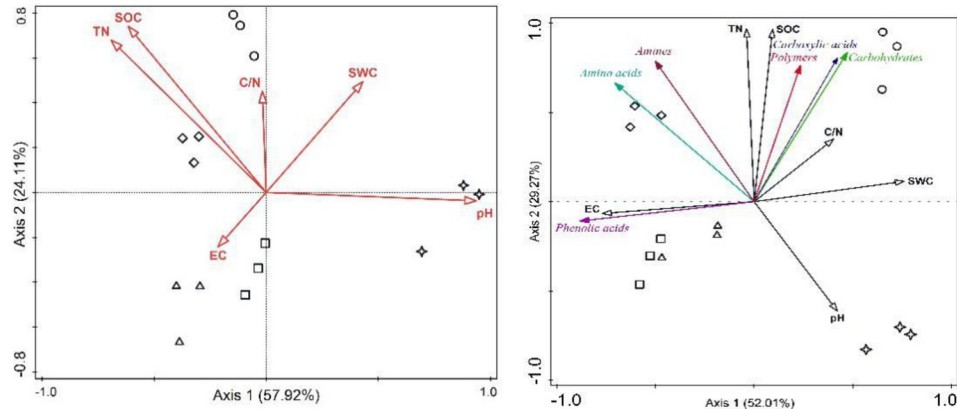

**Fig 5.** RDA based on AWCD and soil properties (left) and six carbon source utilization and soil properties(right) among different extract concentration treatments. Symbols indicate concentrations (circle: 0 g·mL$^{-1}$, triangle: 2.5 mg·mL$^{-1}$, diamond: 5 mg·mL$^{-1}$, square: 7.5 mg·mL$^{-1}$, star: 10 mg·mL$^{-1}$). SWC: soil water content, EC: electrical conductivity, TN: total nitrogen, SOC: soil organic carbon and C/N: C: N ratio.

## Discussion

Our findings of allelopathic effects of invasive species on plant growth and soil microbial activity shed new light on the poorly understood mechanisms underlying the role plant–soil interactions play in shaping invasion success. Invasive species root extracts decrease plant biomass, soil SOC and TN, increase soil EC and pH, and alter microbial carbon source utilization. Taken together, our results indicate how allelopathy impacts both plant and soil community further facilitates the successful invasion.

Consistent with previous studies [23], our results show that root extracts of *R. typhina* may result in dramatic changes in soil community activity by alter soil properties, and in turn to alter the growth of other plants. According to the NWH, invasive species disturb native species growth via root allelopathy [6], altering soil microbial communities [24, 25]. There is increasing evidence that plant–soil interactions can promote successful invasions of alien species and drive plant–plant interactions [5, 26–28]. Chemical allelopathy is suggested as a fundamental mechanism by which invasive plant species displace native ones [12]. Some studies found that successful invaders suppress neighboring plant species growth through allelopathy, via altering soil community's composition [23, 29]. Allelochemicals produced by alien invaders may disrupt regular decomposing processes in soil community when they are released into the novel ecosystems [30]. Our study, conducted in controlled growth chambers, provides evidence that allelopathic effects change with concentration of root extracts. In particular, soil microbial community functional diversity, as characterized by CLPP, is sensitive to root extracts.

The different patterns of microbe carbon utilization indicate the variability in soil microbial metabolic potential in response to root extracts. *R. typhina* roots extracts decrease the AWCD, diversity and evenness, and inhabit on soil microbial carbon utilization activity. AWCD reflects single carbon source utilization ability and metabolic activity of soil microbe, and the lower the value, the lower the metabolic activity of soil microorganism [31, 32]. All six groups of substrate utilization are altered by root extracts, with carbohydrates, carboxylic acids and polymers decreasing and phenolic acid increasing in treated soils. This indicates that soil microbial communities feed less on carbohydrates, carboxylic acids and polymers but more on phenolic acid in soils with root extracts. Hydroxybenzoic and hydroxycinnamic acids are the two major types of phenolic acids produced by *R. typhina*, which may enhance phenolic acid consumption of soil microbial communities.

Our results show that *R. typhina* root extracts increase soil pH and EC, while decrease SOC and TN, in accordance with previous studies [33, 34]. Increase in pH may be caused by a higher uptake of nitrate as the preferential nitrogen source [35, 36]. By altering soil pH and changing the microbial community activity, invasive plants can modify the native plant nutrient uptake [23, 37]. Previous experiments indicate that invasive plants increase utilization of nutrients with allelopathic substance that also inhibit native plant growth [38, 39]. According to this evidence, our study suggests that changes in soil pH, TN, SOC and EC are the main factors shifting soil microbial activity.

In conclusion, our study suggests that *R. typhina* can increase its fitness via allelopathic biochemicals that modify the EC, soil pH, SOC, TN and its microbial community metabolic activity, ultimately inhibiting the growth of *T. erecta*. We highlight that interactions between plants and soil microbes are critical for the successful invasion of alien plant species.

## Acknowledgments

We thank Liang Jin and Jianfeng Zhang for providing valuable comments on this manuscript.

## Author Contributions

**Data curation:** Weiqiang Guo.

**Funding acquisition:** Chunli Zhao, Gianalberto Losapio.

**Methodology:** Yulan Peng.

**Project administration:** Chunli Zhao.

**Resources:** Yulan Peng.

**Writing – original draft:** Tongbao Qu, Xue Du.

**Writing – review & editing:** Tongbao Qu, Gianalberto Losapio.

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
