## [Decision Letter · Decision Letter 0]

3 Nov 2020

PONE-D-20-31348

Invasive species allelopathy decreases plant growth and soil microbial activity

PLOS ONE

Dear Dr. Qu,

Thank you for submitting your manuscript to PLOS ONE. After careful consideration, we feel that it has merit but does not fully meet PLOS ONE’s publication criteria as it currently stands. Therefore, we invite you to submit a revised version of the manuscript that addresses the points raised during the review process.

The manuscript has been reviewed by two referees that raised a series of critical comments. Please prepare a revised version addressing all these comments.

We look forward to receiving your revised manuscript.

Kind regards,

Raffaella Balestrini

Academic Editor

PLOS ONE

Journal Requirements:

Reviewers' comments:

Reviewer's Responses to Questions

**Comments to the Author**

1. Is the manuscript technically sound, and do the data support the conclusions?

Reviewer #1: No

Reviewer #2: Yes

2. Has the statistical analysis been performed appropriately and rigorously? 

Reviewer #1: Yes

Reviewer #2: N/A

3. Have the authors made all data underlying the findings in their manuscript fully available?

Reviewer #1: Yes

Reviewer #2: No

4. Is the manuscript presented in an intelligible fashion and written in standard English?

Reviewer #1: Yes

Reviewer #2: Yes

5. Review Comments to the Author

Reviewer #1: The aim of the work is to demonstrate the allelopathic effect of the plant Rhus typhina on Tagetes erecta and soil microbial population. Authors use a water extract from R. typhina roots. The extract was obtained after grinding the dried roots, incubating the root powder in water for 48 h under shaking, concentrating the extract in a rotary evaporator and freeze drying. The extract was then used to prepare solutions at different concentrations. Solutions were used to water pots containing plants. All the following analyses (plant growth, soil properties, microbial community diversity) were done on soil and plants sampled from these pots.

The work is well described; however, I have a big concern about the pertinence of using the root extract.

Author should explain how this root extract added to the soil can mimic what R. typhina is really releasing in the soil during its growth. In fact, the extract contains the molecules that water could extract from the whole root, but there is no proof that those molecules are the ones that R. typhina roots exudate in the soil. In the introduction, authors cite the “novel weapon hypothesis” and say that “Our results support the ‘novel weapons hypothesis’”. The cited hypothesis is proposed by Callaway and Ridenour (2004) who say “that some exotics transform from native weaklings to invasive bullies by exuding biochemicals that are highly inhibitory (allelopathic) to plants or soil microbes in invaded communities”. The hypothesis is based on molecules released as exudates. Also, the cited works in the discussion are about root exudates, rhizospheric soil, leaf leachates. Authors describe the effects on T. erecta growth and on soil bacteria, but those effect are due to molecules extracted from roots non exudated by roots. There is a big difference between root exudates and root extract.

Also, the choice of the control is not convincing. In my opinion, a more appropriate control would be not water but root extract from a plant that do not have an allelopathic effect on T. erecta, like one historically living in the same habitat. Authors should be sure that the effect on soil microbial activity is due specifically to R. typhina molecules and not to a general shift in microbial population due to the addition to the soil of any root extract.

In some points of the discussion authors over-interpret their data:

lines 202-203: they state “Taken together, our results indicate how allelopathy impacts both plant and soil community, revealing that plant–soil microbe interactions can mediate invasion success”. I agree on the first part on the sentence, since root extract do have an effect on plant growth and on microbial population. However, data do not help us understanding whether the effect on plant is direct or mediated by microbes.

Line 204: “our results show that plants have the potential to alter soil”: this sentence is misleading: written like this means that plants, in their active growth, can alter soil, but results are on root extract not on exudates.

Lines 206-207. “This study suggests that invasive species can increase their own fitness relative to other plant species by modifying soil conditions and microbial community” and Lines 238-239 “are the main factors shifting soil microbial activity and mediating the impact of invasive plants on the growth of other plants.” Again, there is not proof in the work that effect on plants is either direct or mediated by other factors.

Citation of literature is sometimes not appropriate

Line 209: Tian et al 2017: this work is about the effect of N fertilisation on soil microbial community, not on allelopathy

Lines 226-228: “These results are partially in agreement with previous studies indicating that root extracts serve as soil carbon substrates for specific soil microbial communities (Fan et al. 2010, Wu et al. 2013, Li et al. 2017)”. None of these cited papers deal about root extracts: they describe works on exudates and rhizospheric soil.

Minor points

Line 41: either remove “is”, or add “which” before “is”

Line 64: Substitute “As” with “It is” or rephrase the sentence

Line 98: “five plant” modify in “five plants”

Line 137: remove “in”

Line 138: change “concentration extracts” to “extract concentrations”

Lines 146-147: according to Table 1, plant height is not significantly different from control at 10 mg/mL.

Line 164: table 2 shows soil properties data not AWCD.

Line 230: Olchowik et al. 2012 demonstrated the presence of tannins in leaves: there is no indication of their presence in roots

Reviewer #2: The manuscript Invasive species allelopathy decreases plant growth and soil microbial activity aims at demonstrating the capaicty of the invasive shurb R. typhina to inihibit other plants growth to invade the soil. Data reported support the hypothesis that R. typhina roots extracts inhibited T. erecta growth via altering the physico-chemical soil conditions, as well as the microbial community.

The manuscript is well written and presented in intellegible fashion. Beside I am not English mother-tongue, it seems that the manuscript has been written in standard English.

The conclusion are supported by the reported data.

However more information on the conditions in which T. erecta has been grown during the experimental condition,such as light humidity, temperature, photoperiod.

More information about the use Community-level physiological profiles (CLPP) should be provided for reader who are not familiar with such method.

6. PLOS authors have the option to publish the peer review history of their article (what does this mean?). If published, this will include your full peer review and any attached files.

Reviewer #1: No

Reviewer #2: **Yes: **Biancaelena Maserti

---

## [Author Response · Author response to Decision Letter 0]

5 Dec 2020

1.We revise the manuscript and ensure that your manuscript meets PLOS ONE's style requirements, including those for file naming.

2.All relevant data are within the paper.

3.We amend our list of authors on the manuscript to ensure that each author is linked to an affiliation.

---

## [Decision Letter · Decision Letter 1]

25 Jan 2021

Invasive species allelopathy decreases plant growth and soil microbial activity

PONE-D-20-31348R1

Dear Dr. Qu,

We’re pleased to inform you that your manuscript has been judged scientifically suitable for publication and will be formally accepted for publication once it meets all outstanding technical requirements.

Kind regards,

Raffaella Balestrini

Academic Editor

PLOS ONE

Additional Editor Comments (optional):

Reviewers' comments:

Reviewer's Responses to Questions

**Comments to the Author**

1. If the authors have adequately addressed your comments raised in a previous round of review and you feel that this manuscript is now acceptable for publication, you may indicate that here to bypass the “Comments to the Author” section, enter your conflict of interest statement in the “Confidential to Editor” section, and submit your "Accept" recommendation.

Reviewer #1: All comments have been addressed

Reviewer #2: All comments have been addressed

2. Is the manuscript technically sound, and do the data support the conclusions?

Reviewer #1: (No Response)

Reviewer #2: Yes

3. Has the statistical analysis been performed appropriately and rigorously? 

Reviewer #1: (No Response)

Reviewer #2: Yes

4. Have the authors made all data underlying the findings in their manuscript fully available?

Reviewer #1: (No Response)

Reviewer #2: Yes

5. Is the manuscript presented in an intelligible fashion and written in standard English?

Reviewer #1: (No Response)

Reviewer #2: Yes

6. Review Comments to the Author

Reviewer #1: Authors took in consideration all my concerns and replied accordingly.

Minor points

lines 25 and 209: change "in particularly" with "in particular"

Reviewer #2: The authors responded to all my previous comments and now the manuscript could be published.

I reccomen to change to lower case, the initial character of roots at line 22: roots, instead Roots.

7. PLOS authors have the option to publish the peer review history of their article (what does this mean?). If published, this will include your full peer review and any attached files.

Reviewer #1: No

Reviewer #2: **Yes: **Biancaelena Maserti

---

## [Editor Report · Acceptance letter]

27 Jan 2021

PONE-D-20-31348R1 

Invasive species allelopathy decreases plant growth and soil microbial activity 

Dear Dr. Qu:

I'm pleased to inform you that your manuscript has been deemed suitable for publication in PLOS ONE. Congratulations! Your manuscript is now with our production department. 

Kind regards, 

on behalf of

Dr Raffaella Balestrini 

Academic Editor

PLOS ONE